# Advances in Genetic Engineering in Improving Photosynthesis and Microalgal Productivity

**DOI:** 10.3390/ijms24031898

**Published:** 2023-01-18

**Authors:** Jinlu Hu, Dan Wang, Hui Chen, Qiang Wang

**Affiliations:** 1School of Life Sciences, Northwestern Polytechnical University, Xi’an 710072, China; 2State Key Laboratory of Crop Stress Adaptation and Improvement, School of Life Sciences, Henan University, Kaifeng 475004, China; 3Academy for Advanced Interdisciplinary Studies, Henan University, Kaifeng 475004, China

**Keywords:** microalgae, photosynthesis, ncRNAs, biomass production, genetic engineering

## Abstract

Even though sunlight energy far outweighs the energy required by human activities, its utilization is a key goal in the field of renewable energies. Microalgae have emerged as a promising new and sustainable feedstock for meeting rising food and feed demand. Because traditional methods of microalgal improvement are likely to have reached their limits, genetic engineering is expected to allow for further increases in the photosynthesis and productivity of microalgae. Understanding the mechanisms that control photosynthesis will enable researchers to identify targets for genetic engineering and, in the end, increase biomass yield, offsetting the costs of cultivation systems and downstream biomass processing. This review describes the molecular events that happen during photosynthesis and microalgal productivity through genetic engineering and discusses future strategies and the limitations of genetic engineering in microalgal productivity. We highlight the major achievements in manipulating the fundamental mechanisms of microalgal photosynthesis and biomass production, as well as promising approaches for making significant contributions to upcoming microalgal-based biotechnology.

## 1. Introduction

With a growing global population, dwindling agricultural land and climate change, there is a strong demand for more productive and stress-resistant crops for food and energy purposes. Among photosynthetic organisms, microalgae and cyanobacteria are the most promising feedstocks for meeting the rising demand for food, feed, fuel, and high-value metabolites owing to their higher growth rates than those of terrestrial crop plants; additionally, they can be used for wastewater treatment and CO_2_-emissions mitigation processes [1]. However, microalgae-based products are currently hampered by high production costs and inefficient light use. Photosynthesis is the primary driving force behind microalgal growth and biomass production, as it provides the energy and carbon requisite for the biosynthesis of organic compounds [2]. To increase growth rates and microalgal productivity and thus make the process profitable, light-to-biomass conversion efficiency must be optimized [1]. 

Because energy losses happen at numerous stages, even during the light-driven conversion of CO_2_ to organic carbon, the overall efficiency of sunlight-to-biomass conversion reported in mass cultures is much lower (35–80%) than the predicted theoretical maximum [3]. Further increases in microalgal productivity are expected to be achieved through genetic engineering. The light-harvesting system and CO_2_ assimilation are two major sources of energy losses and, thus, foremost goals for genetic engineering. Several strategies for improving photosynthesis capacity, focusing on the light and dark phases of photosynthesis, have been presented. Another strategy is to manipulate non-coding RNAs (ncRNAs) and transcription factors (TFs) to regulate photosynthesis. These strategies have a good effect on improving microalgal photosynthesis and increasing multiple metabolites. The purpose of this review is to elaborate on the key factors for photosynthesis and some important products throughout the conversion of sunlight into biomass and to sum up previous efforts taken aimed at increasing photosynthetic efficiency and microalgal productivity through genetic engineering.

## 2. Strategies for Increasing Photosynthetic Efficiency

Photoautotrophs convert sunlight into organic molecules and biomass through oxygenic photosynthesis, albeit the efficiency of this process varies depending on the species and environmental factors [4]. Oxygenic photosynthesis begins with light absorption, followed by excitation energy transfer to the reaction centers, primary photochemistry, electron and proton transport, NADPH and ATP synthesis, and then CO_2_ fixation (Calvin-Benson cycle) [5]. So far, multiple techniques have been applied to increase the efficiency of light-to-biomass conversions, such as optimization of culture conditions, mutagenesis, mutant selection, and genetic engineering [6,7,8]. The application of modern molecular tools and genetic engineering facilitates the construction of directed phenotypic mutant strains. Changing the expression of key genes in the photosynthetic metabolic pathway can effectively improve photosynthetic efficiency and biomass accumulation. In addition, non-coding RNAs (ncRNAs) and transcription factors (TFs) associated with photosynthesis are also valuable regulators (Figure 1). In the future, more paradigms are needed for the improvement of microalgae that afford higher efficiency.

### 2.1. The Light Phase of Photosynthesis

Both land plants and microalgae have relatively inefficient photosynthesis. In the early stages of light collection, approximately 75% of the energy generated by solar irradiation is lost because not all of the light spectrum is used; some are reflected or transmitted, and some is wasted as heat. Even if some energy loss is unavoidable or required for photoprotection, there are nevertheless aspects in which light reaction efficiency could be enhanced, potentially leading to a significant increase in crop yields [9]. Some of the ideas focus on increasing light absorption or accelerating the photosynthetic electron transport pathway. 

Over 50% of the energy losses associated with the conversion of solar energy into chemical energy during photosynthesis are attributed to kinetic restrictions between the fast rate of photon capture by the light-harvesting apparatus and the slower downstream rate of photosynthetic electron transfer [10]. Optimizing light collection and use by minimizing chlorophyll antenna size is one strategy for increasing energy conversion efficiency and photosynthetic productivity [11]. Theoretically, in mass cultures of algae or plants, truncated photosystem chlorophyll antenna size can increase photosynthetic solar energy conversion efficiency and productivity by up to thrice [12]. In all types of photosynthetic organisms, a shortened light-harvesting chlorophyll antenna size (TLA) would help to reduce excessive sunlight absorption and the following wasteful non-photochemical dissipation of excitation energy [13,14]. DNA insertional mutagenesis experiments were the first to demonstrate that a truncated chlorophyll antenna would result in relatively higher photosynthetic productivity in the model organism *Chlamydomonas reinhardtii* (henceforth *C. reinhardtii*) [14,15]. Strains *tla3* and *tla4* in *C. reinhardtii* were mutated in these genes encoding the chloroplast-localized signal recognition particle (CpSRP) and showed increased efficiency of solar energy conversion and photosynthetic productivity in mass culture under strong irradiation conditions [16,17]. Components of the CpSRP complex are interesting molecular targets for shrinking the Chl antenna without compromising photosynthetic electron transport, which is involved in the appropriate folding of Light-harvesting complex proteins (LHCs) and being targeted to the thylakoids [13]. The BF4 and *p71* antenna mutants from *C. reinhardtii* have defects in the insertase Alb3.1 and cpSRP43, resulting in a truncated antenna size in the two photosystems [18] and impaired accumulation of LHCs [19]. Furthermore, CRISPR-Cas9 technology has recently been demonstrated to be a reliable approach for producing *tla* mutants [20,21]. *CAO*, which encodes for Chlorophyllide, an oxygenase responsible for Chl *a* to Chl *b* conversion, was another molecular target that was expected to influence antenna size [22]. At high light intensities, a *CAO* mutant modulated by RNAi with decreased chlorophyll *b* increased its photosynthetic rate by more than twofold [10]. By expressing a *CAO* gene with a 5′ mRNA extension encoding a Nab1 translational repressor binding site in a *CAO* knockout line, the mutant having light-regulated antenna sizes had substantially higher photosynthetic rates and two-fold greater biomass productivity than the parental wild-type strains [23]. In addition, a phycocyanin-deletion (Δ*cpc*) mutant of the cyanobacterium *Synechocystis* sp. PCC 6803 (henceforth *Synechocystis* 6803) demonstrated that biomass accumulation was 1.57 times larger than that of the WT under strong light and high cell density conditions, with a lower Chl per cell content and a lower PSI/PSII reaction center ratio than the WT [24]. Truncated antenna mutants of *Chlorella sorokiniana* (henceforth *C. sorokiniana*), with a 30–68% higher biomass yield in dense cell suspensions typical of industrial photobioreactors, showed increased photon use efficiency and higher productivity compared to WT [25,26]. In the marine diatom *Phaeodactylum tricornutum* (*P. tricornutum*), knockout of ALB3b, which is involved in the regulation of fucoxanthin-chlorophyll *a/c* synthesis, exhibits a truncated light-harvesting antenna phenotype with lower levels of photosynthetic pigments [27]. The *alb3b* mutants displayed 30–40% higher rETRmax (the maximum relative electron transport rate) and Ek (light saturation index) compared with the wild-type from low light (35 μmol photons m^−2^ s^−1^) to medium light (200 μmol photons m^−2^ s^−1^), showing the cells had been able to downsize the photosynthetic apparatus in response to the increased light intensities.

The prospect of creating engineered microalgae with light-harvesting systems that do not exist in nature or wild-type cells is a more radical method for enhancing light-energy consumption. Usually, only the visible portion of the solar spectrum (from 400 to 700 nm) is used for photosynthesis due to the spectral properties of photosynthetic pigments. Small modifications or variations of chlorophyll enable photosynthetic organisms to absorb sunlight at various wavelengths [28]. Chlorophyll *d* [29,30] and chlorophyll *f* [31] have been shown to use far-red light (FRL; from 700 to 750 nm) in some cyanobacteria to perform photosynthesis. The overexpression of endogenous CBPII (chlorophyll *d*-binding light-harvesting protein) from *Acaryochloris marina* suppressed the phycobiliproteins of *Synechocystis* 6803, resulting in a low ratio of phycobilins to chlorophyll *a* [29]. Transplastomic algae expressing the Katushka fluorescent protein increased oxygen evolution and photosynthetic growth in yellow light and enhanced the photosynthetic action spectrum of *C. reinhardtii* [32]. Introducing alternative light-harvesting complexes that absorb more efficiently in areas where chlorophyll is less efficient, like chlorophyll *f* [31], chlorophyll *d*-binding light-harvesting proteins [29,30], engineered fluorescence proteins [32], or diatom fucoxanthin components [33] is a possibility. Although this idea is plausible, whether it is feasible is a long-term question.

Non-photochemical quenching (NPQ) is an important photoprotective molecular mechanism inducing the thermal dissipation of absorbed light energy in oxygenic photosynthetic organisms [34]. There is little doubt that most vegetation on the planet will use NPQ on a daily basis, and it represents a pathway for our biosphere to process substantial quantities of solar energy. Although this protective dissipation is indispensable, it keeps operating even when high to normal/low light transition, which minimizes photosynthetic efficiency [35]. Hence, tuning of NPQ has been reported as a promising biotechnological strategy for increasing biomass productivity in microalgae. Two light-harvesting complex stress-related proteins, LHCSR1 and LHCSR3, were reported as the main actors during NPQ induction in *C. reinhardtii* [36,37]. A mutant of *C. reinhardtii* lacking LHCSR, *npq4lhcsr1*, displays high rates of photosynthesis when grown in high light compared to the wild-type [38]. The strain lacking LHCSR1 and knocked down in LHCSR3, causing enhanced singlet oxygen release and PSII photodamage, had an improved photosynthetic efficiency under high light [39]. It is noteworthy that faster NPQ relaxation and improved crop photosynthetic efficiency can be achieved under fluctuating light conditions by overexpressing NPQ-related genes [40,41]. The concept may be worth further verification and exploration to develop stress-resilient microalgae with higher photosynthetic output.

### 2.2. The Carbon Reactions of Photosynthesis

Central to many strategies to improve photosynthetic efficiency is addressing the limitation of RuBisCO, a rate-limiting enzyme in photosynthesis [42]. Plants produce a large amount of RuBisCO to compensate for its low activity, accounting for up to 50% of photosynthetic organisms’ soluble proteins. However, this necessitates a significant nitrogen investment in RuBisCO. As a result, engineering microalgal strains with enhanced RuBisCO catalytic activity would be critical for improving solar energy conversion efficiency. There are currently three approaches to genetically engineering carbon fixation in microalgae: (1) endogenous overexpression, (2) site-directed mutagenesis, and (3) RuBisCO isoforms.

Endogenous overexpression and site-directed mutagenesis were used to create some RuBisCO-improved variants by targeting either the *rbcL* and *rbcS* genes (RuBisCO subunits) or the subunit that interacts with RuBisCO activase [43,44]. The overexpression of endogenous RuBisCO activase enhanced lipid and biomass efficiency by up to 40% in *Nannochloropsis oceanica* [43]. Consistently, overexpression of RuBisCO in *Synechocystis* 6803 improved photosynthetic activity and fatty acid productivity [45,46,47,48]. The site-directed mutagenesis of *rbcL* produced a low-activity RuBisCO variant that induced higher hydrogen production rates and total lipid levels in *C. reinhardtii* than in the wild type [44,49]. The engineering of microalgal strains with hybrid RuBisCO complexes would also be crucial to improving RuBisCO catalytic activity and the efficiency of solar energy conversion. Combining positive mutations from different isoforms has been proposed as a method of obtaining RuBisCO with improved carboxylation catalysis V_max_ [50,51]. To increase the CO_2_/O_2_ selectivity and carboxylation catalytic efficiency, the small subunit of the RuBisCO enzyme of *C. reinhardtii* was swapped out for those from Arabidopsis, spinach, and sunflower in one such endeavor [51]. The pyrenoid is a subcellular microcompartment in which algae sequester Rubisco, thus realizing the CO_2_-concentrating mechanism (CCM) [52]. Since the algal CCM is functionally analogous to the terrestrial C_4_ pathway in higher plants [53], these findings could pave the way for transforming algae and achieving higher productivity.

In addition to RuBisCO, the other relatively low-abundant enzymes in the Calvin-Benson cycle, such as sedoheptulose-1,7-bisphosphatase (SBPase), fructose-1,6-bisphosphatase (FBPase), and fructose-1,6-bisphosphate aldolase (FBA), are the prime targets to control the photosynthetic efficiency. The engineering of the Calvin-Benson cycle through the overexpression of cyanobacterial FBA was shown to improve the cell growth and photosynthetic activity of *Chlorella vulgaris* (*C. vulgaris*) [54]. Similarly, it was discovered that overexpression of cyanobacterial FBP/SBPase increased photosynthetic activity in *Euglena gracilis* [55]. The overexpression of *C. reinhardtii* SBPase was promoted to improve photosynthetic capacity, total organic carbon content and osmoticum glycerol production in *Dunaliella bardawil* [56]. Endogenous overexpression of RuBisCO, FBA, and SBPase increased oxygen evolution in vivo and biomass accumulation in *Synechocystis* 6803 [57], significantly increasing the generation of ethanol [58]. Therefore, engineering key enzymes of the Calvin-Benson cycle continues to be a promising target for increasing photosynthetic efficiency.

### 2.3. Non-Coding RNAs and Transcription Factors Affecting Photosynthesis

ncRNAs are transcriptional and posttranscriptional regulators of gene expression that play important roles in almost every aspect of an organism’s life cycle [59]. Complex sets of endogenous ncRNAs, including candidate microRNAs (miRNAs) and small RNAs (sRNAs), have now been identified by high-throughput sequencing and experimental validation in eukaryotic algae and cyanobacteria. Several ncRNAs play critical roles in the acclimation to environmental changes relevant to oxygenic photosynthesis in cyanobacteria, especially *Synechocystis* 6803 (Table 1). sRNA ApcZ links the expression of the *apcABC* operon that encodes the Apc core proteins of the PBS, providing a functional and mechanistic link between light harvesting and photoprotection [60]. IsaR1 is widely conserved in the cyanobacterial phylum, including freshwater, marine, filamentous, symbiotic, mesophilic, or thermophilic cyanobacteria [61]. IsaR1 controls a complex network important for iron acclimation and acts on the photosynthetic apparatus in three distinct ways, involving the major ferredoxin Fed1 (*petF*), cytochrome c6 (*petJ*), the cytochrome b6f complex proteins PetABDC1, glutamyl-tRNA reductase (*hemA*), and the biosynthesis of iron-sulfur clusters (*sufBCDS*). PsrR1 has been found to be widely conserved in cyanobacteria and limits the expression of photosynthesis-related genes (*psaL*, *psaJ*, *chlN*, *cpcA*, and several others) upon shift to a high light [62]. PsrR1 transcription is upregulated at higher light levels to achieve this regulation. The above upregulation is mediated by the response regulator RpaB, which loses its ability to bind DNA when it switches to HL, resulting in a rapid de-repression of *psrR1* transcription within minutes [63]. 

Cis-encoded antisense sRNAs (asRNAs) that are located on the opposite strand of DNA from their mRNA targets have a high complementarity to their targets. In *Synechocystis* 6803, three asRNAs, RblR, PsbA2R, and PsbA3R, regulate photosynthesis by positively modulating their respective targets, the *rbcL*, *psbA2*, and *psbA3* mRNAs [64,65]. RblR acts as a positively acting factor to regulate the *rbcL* gene expression under multiple stress conditions [64]. In addition, RbcR, as a RuBisCO regulator, binds the *rbcL* promoter and affects the expression of several genes involved in C acquisition, including *rbcLXS*, *sbtA*, and *ccmKL*, which encode RuBisCO and parts of the CCM, respectively [66]. Overexpression of PsbA2R increased the amount of *psbA*-encoded D1 protein and the potential for photosynthetic activity under high light conditions by protecting an RNase E-sensitive region [65]. AsRNA As1-Flv4 prevents premature expression of the flv4-2 operon, providing many β-cyanobacteria with a previously unknown photoprotection mechanism that evolved in parallel with oxygen-evolving PSII after the shift to inorganic carbon via co-degradation [67,68]. IsrR is the first known to regulate a photosynthesis component and is a repressor of the iron stress-induced protein IsiA, which forms a giant ring structure around PSI [69]. 

More and more ncRNAs and transcription factors related to photosynthesis are being discovered and studied in depth. In the future, using ncRNAs and transcription factors to improve the photosynthesis and biomass of microalgae is a promising paradigm. 

**Table 1 ijms-24-01898-t001:** ncRNAs relating to photosynthesis in cyanobacteria.

Name	Type	Length	Species	Function	Reference
ApcZ	sRNA	137	*Synechocystis* 6803	Inhibiting *ocp* translation under stress-free conditions	[60]
IsaR1	sRNA	68	Conserved in cyanobacteria	Limiting photosynthesis-related gene expression (*petJ*, *petABDC1*, *hemA*, *sufBCDS*, and several others) under low iron conditions	[61]
PsrR1	sRNA	131	Conserved in cyanobacteria	Limiting photosynthesis-related gene expression (*psaL*, *psaJ*, *chlN*, *cpcA*, and several others) upon shift to HL	[62]
RblR	asRNA	113	*Synechocystis* 6803	Activating *rbcL* expression	[64]
PsbA2R PsbA3R	asRNA	130, 220 160, 180	*Synechocystis* 6803	Protecting *psbA2* and *psbA3* mRNA from premature degradation	[65]
As1-Flv4	asRNA	280, 500	*Synechocystis* 6803	Preventing premature expression of the *flv4-2* operon after shift to LC	[67]
IsrR	asRNA	177	*Synechocystis* 6803	Inhibiting *isiA* expression under iron stress	[69]

## 3. Transgenic Microalgae for Improved Biomass Production

Solar energy and carbon dioxide can be converted into commercially valuable organic compounds such as polyunsaturated fatty acids (PUFAs), pigments, proteins, and polysaccharides by microalgae. Furthermore, the cultivation of microalgae does not fight with agricultural food production and can be grown in marine environments such as freshwater, seawater, or even wastewater, making them promising biocatalysts for applications in sustainable food, fuel, and chemical production. Novel genome editing tools such as RNAi, CRISPR/Cas9, ZNFs, and TALENs have been used in recent years to improve the quality and quantity of desired products. In addition, genetic engineering is frequently used because they produce faster and more precise results than random mutagenesis [70]. 

The advancement of genetic engineering, transcriptional engineering, and metabolic engineering strategies has resulted in breakthroughs in research on functional characterization of key genes or regulators, identification of metabolic pathways, and elucidation of microalgae cell physiology [71]. In this part, we will describe the synthetic pathways for high-value bioproducts derived from microalgae, as well as strategies for increasing bioproduct accumulation (mainly lipids, pigments, and polysaccharides).

### 3.1. Lipids

Though the research on microalgal lipids is still in its infancy, the pathway of triacylglycerol (TAG) synthesis in microalgae is very similar to that of higher plants. Thus, it provides a relatively mature framework for the initial study of metabolic pathways in microalgae [72]. The synthesis of TAG in microalgae can be divided into the fatty acid synthesis pathway and the Kennedy pathway. Regulation of enzymes in the fatty acid synthesis pathway and Kennedy pathway is a breakthrough to improve lipid accumulation in microalgae. 

The key enzymes in the fatty acid synthesis pathway are pyruvate dehydrogenase (PDH), acetyl-CoA carboxylase (ACCase), and acetyl-CoA [73]. The neutral lipid content of the PtPDK antisense knockdown mutant strain increased by up to 82%, while fatty acid composition remained unchanged in *P. tricornutum* [74]. The results showed that acetyl-CoA can be generated from pyruvate via PDH and is negatively regulated by PDH kinase (PDK). NsPDK knockdown via RNAi altered the fatty acid profile in *Nannochloropsis salina*, leading to faster TAG accumulation without compromising cell growth under high light stress conditions [75]. ACCase, which is the first pivotal enzyme in microalgal lipid synthesis and catalyzes the rate-limiting step for fatty acid biosynthesis, has attracted the attention of many scholars. The ACCase inhibitors resulted in a marked decrease in TAG accumulation levels, but ACCase overexpression caused no significant changes in microalgal lipid accumulation [76]. 

Compared to the fatty acid synthesis pathway, the Kennedy pathway is relatively closer to the target product and, therefore, more likely to influence TAG synthesis. The Kennedy pathway mainly contains three acyltransferases, i.e., glycerol-3-phosphate acyltransferase (GPAT), diacylglycerol acyltransferase (DGAT), and lysophosphatidic acid acyltransferase (LPAT), which catalyze the specific esterification of glycerol-3-phosphate (G3P) [77]. Niu et al. studied the effect of GPAT overexpression in *P. tricornutum* on lipid accumulation. These results showed that the neutral lipid content was enhanced twofold and the fatty acid composition had a significantly higher proportion of unsaturated fatty acids in the GPAT overexpression mutant strain compared to the wild type [78]. AGPAT1 overexpression in *P. tricornutum* coordinated the expression of other key genes associated with TAG syntheses, such as DGAT2 and GPAT, and enhanced TAG content by 1.81-fold with a significant increase in polyunsaturated fatty acids, primarily EPA and DHA, and yet reduced the content of soluble proteins and total carbohydrates [79]. Overexpression of NeoLPAAT1 in *Neochloris oleoabundans* increased total lipid content and TAG content by twofold compared to the wild type [80]. In addition to overexpressing single or multiple target genes, Zou et al. attempted to design a strong constitutive promoter Pt211 to increase the expression level of multiple target genes in *P. tricornutum* [81]. The qPCR analysis showed that GUS, GPAT and DGAT2 genes involved in TAG biosynthesis showed higher transcript abundances, while algal growth and photosynthesis were not impaired. 

Carbonic anhydrase (CA) are widespread enzymes that catalyze CO_2_ hydration to bicarbonate, which is essential for the carbon-concentrating mechanism in microalgae [82]. The heterologous CA from *Sulfurihydrogenibium yellowstonense* (SyCA) and *Mesorhizobium loti* (MlCA) were explored to increase CO_2_ capture and utilization using various culture devices in *C. reinhardtii*. Moreover, the biomass, lutein, and lipids were increased 2-, 4-, and 8-fold in genetically modified *C. reinhardtii* [82]. The genetically engineered algae harboring exogenous MlCA had improved biomass production, protein content and lipid accumulation in *C. sorokiniana* and *C. vulgaris*. The results showed that the transformants produced up to 1.1 g/L of lipid, which was 2.2-fold higher than the wild types, even while boosting carbon capture and fixation [83]. Although this step is not strictly part of the fatty acid synthesis pathway or the Kennedy pathway, it provides a new idea that increasing the content of important precursors in the pathway can also increase lipid accumulation.

### 3.2. Pigments

Microalgae are a powerful, promising, renewable, and high-quality source of biopigments [84]. It is feasible to generate pigments such as chlorophyll, β-carotene, lutein, zeaxanthin, phycobiliproteins, and lycopene. Pigments have potential health benefits and are used in the treatment and prevention of a variety of diseases. For example, phycobiliproteins have been associated with antioxidant, anticancer, and anti-inflammatory capacities [85]. As shown in Table 2, all of these pigments have potential applications.

Because of their numerous health and industrial applications, microalgal carotenoids are the most commercially produced natural pigments. Because microalgae can synthesize a wide range of carotenoid species, determining metabolic pathways is an important step before engineering algal strains for industrial applications. The biosynthesis of carotenoids may differ between species, but they all have a common metabolic pathway. IPP or DMAPP, a five-carbon precursor, is synthesized through the methylerythritol 4-phosphate (MEP) pathway from pyruvate and glyceraldehydes-3-phosphate, and condensation of such C5 units produces different C10, C15, and C20 polyprenyl units, one of which is geranylgeranyl pyrophosphate (GGPP) [100]. Phytoene synthase (PSY) is an enzyme that catalyzes the reaction of two GGPP molecules to form a 40-carbon phytoene, which is the first limited step in carotenoid biosynthesis and a common precursor of other carotenoids in microalgae [101,102]. Phytoene is converted to lycopene by carotenoid isomerase (CRITISO), ζ-carotene desaturase (ZDS), and phytoene desaturase (PDS). The pathway splits into two branches after lycopene. In one of these branches, lycopene is cyclized into β-carotene by lycopene β-cyclase. Carotene-hydroxylase then hydroxylated β-carotene to zeaxanthin, which zeaxanthin epoxidase (ZEP) then epoxidized to violaxanthin. Astaxanthin is a unique carotenoid because it contains oxygen in both oxy- and hydroxyl groups. β-carotene ketolase (BKT) usually converts zeaxanthin or violaxanthin produced from β-carotene into astaxanthin [103].α-carotene is produced in the other branch via coordinated catalysis by ε-cyclases and β-cyclase. Lutein is formed when carotene ε-hydroxylase and carotene β-hydroxylase hydroxylate α-carotene [100]. 

Both genetic and metabolic engineering are effective approaches for increasing pigment production. It provides the necessary access to increase the activity of numerous rate-limiting enzymes through overexpression, resulting in increased productivity. Overexpression of astaxanthin synthase (*crtS*) increased astaxanthin production by 33% through activity and upregulated carotenoid pathway genes [104]. The optimized *Cr*BKT significantly increased the accumulation of astaxanthin and ketocarotenoids [105], thereby enhancing highlight tolerance and productivity in *C. reinhardtii* [106]. In addition, the down-regulation of specific enzymes can be beneficial for carotenoid overproduction by increasing the desired flux while decreasing the flux towards the other branches. Down-regulation of ε-cyclases, for example, is needed for β-carotene overproduction such that lycopene is not transformed into α-carotene [100]. Overexpression of foreign genes can also enhance carotenoid production. The carotenogenic pathway of *Dunaliella salina* (*D. salina*) was metabolically engineered for the production of astaxanthin by incorporating the *bkt* gene encoding BKT from *Haematococcus pluvialis* (*H. pluvialis*) and chloroplast targeting [107]. Because of its ease of use, the CRISPR-Cas9 system is now a widely used technology for genome editing [108]. In *C. reinhardtii* strain CC-4349, the zeaxanthin content of a knockout mutant of the ZEP-encoding gene induced by preassembled DNA-free CRISPR-Cas9 ribonucleoproteins was markedly greater than the wild type [109]. Genome editing using CRISPR-Cas9 is also possible in *C. reinhardtii* [110].

Although some progress has been made in the genetic and metabolic engineering of pigment genes in microalgae, much more research is needed to achieve high productivity. Furthermore, a more clear understanding of algal pigment regulation and its interaction with other metabolic processes is essential for effective algae engineering. For example, many commercially viable bypass compounds, such as terpenoids, have been synthesized from IPP and DMAPP as generic precursors during the pigment synthesis process [111]. Isoprene, the main component of synthetic and natural rubber, is one of the most basic. Heterologous overexpression of *fni*, an isopentenyl isomerase from *Streptococcus pneumonia*, enhanced DMAPP substrate availability and isoprene synthase concentration in *Synechocystis* 6803, resulting in a higher isoprene-to-biomass production ratio [112]. A phycocuanin-phellandrene synthase fusion mutant increased the rates and yield of *β*-phellandrene hydrocarbons production [113]. What is noteworthy is that terpene production will necessarily compete with pigment synthesis as the precursor pathways are the same [111]. To further increase pigment production in microalgae, the carbon fluxes between pigments and bypass products should also be balanced.

### 3.3. Polysaccharides

Microalgae produce polysaccharidic mucilage to protect their cells from desiccation and a variety of extreme fluctuations in environmental conditions such as pH, temperature, salinity, irradiance, and even predators [114]. Polysaccharides (PS) were discovered in microalgae as cell wall components, with one part found in cells peripheral to the glycocalyx or one of the exopolysaccharides (EPS) [115]. Fucoidans, exopolysaccharides, alginates, and carrageenans are an example of microalgal PS. Due to their diverse roles and potential applications for the pharmacological, therapeutic, regenerative medicine, mechanical, and food producers, PS has been the focus of recent and intensive research [116]. For instance, the carbohydrate content of microalgal biomass is used as a feedstock for the generation of organic acids and bioethanol in the fermentative technique [117].

Except for cyanobacteria, which are cytoplasmic, PS biosynthesis and their sulfation in microalgae happened in the Golgi complex [118]. Moreover, PS production occurs primarily during the stationary phase of the microalgae culture [116]. *Chlorella* sp., *Arthrospira platensis*, *Porphyridium*, *D. salina*, and *Euglena gracilis* are the most extensively investigated eukaryotic microalgae and cyanobacteria used for the production and extraction of PS [119,120,121,122,123]. Some such species can generate substantial amounts of EPS during algal cultivation owing to typical physiological processes, whilst others must be stressed in order to synthesize these compounds [114]. For example, to produce exocellular polysaccharides, the red marine microalga required specific growth conditions such as specific N/P ratios or nitrogen starvation [124]. 

Several studies have been conducted to improve glycogen production through the genetic engineering of microalgae in order to maximize EPS productivity. The glycogen biosynthesis in *Synechocystis* 6803 is altered by the depletion of *glgP*, which results in a two-fold increase in glycogen under mixotrophic conditions, indicating that blocking glycogen degradation causes an increased glycogen accumulation [125]. The intracellular glycogen content of a GAP1 gene-overexpressing *Synechococcus* 7002 was found to be 1.2-fold higher than that of the WT, indicating that glycolysis activation promotes glycogen accumulation [126]. CmGLG1 is a glycogenin that is essential for the start of glycogen/starch synthesis in the red alga *Cyanidioschyzon merolae* (*C. merolae*) [127]. The overexpression of CmGLG1 resulted in 4.7-fold higher starch content than the WT. CmGLG2 is another glycogenin involved in the synthesis of floridean starch, as the overexpression of CmGLG2 caused a two-fold increase in floridean starch content in *C. merolae* [128].

However, microalgal polysaccharides are not adapted to recovery during upstream and downstream processes. They are commonly regarded as byproducts of pigment and lipid production. In addition, their high level of structural complexity further reduces their value as high-value molecules [116]. As a result, increasing the yield of microalgal polysaccharide through genetic engineering remains a difficult problem.

## 4. The Limitations and Future Strategies of Genetic Engineering in Microalgal Productivity

Cell metabolism determines a cell’s potential, and genetic and metabolic engineering are key modern technologies for developing a cell into a cell factory [7]. Nowadays, genetic engineering advances allow for the engineering of algal strains to improve both biomass productivity and the yield of high-value products from microalgae [2]. The biosynthesis of carbohydrates, lipids, proteins, and pigments in microalgal cells is highly interrelated in the metabolic network and controlled by limiting steps. The specific cultivation conditions can be set by genetic engineering to shift metabolic fluxes toward different metabolites. To increase the concentration of products and decrease the unitary cost of algal biomass, the major efforts are focused on understanding the metabolic reactions of primary production and obtaining strains with higher photosynthetic efficiency. However, several challenges need to be addressed to achieve the goal of comprehensive utilization of microalgae for enhanced production of multiple compounds. 

First, the genetic and biochemical pathways in microalgae remain unclear. Furthermore, molecular modifications and practical cultivation issues, such as *H. pluvialis*’ sensitivity to environmental vibrations [129], should be thoroughly considered. Second, novel gene editing tools such as RNAi, CRISPR/Cas9, ZNFs, and TALENs have been used to boost byproduct accumulation [130]. However, when microalgae are compared to simple organisms such as bacteria, genome manipulation is still challenging. Third, it is difficult to find low-cost and green techniques for extracting all of the bioactive components of microalgal cells [131]. Furthermore, there are some environmental and economic bottlenecks to large-scale applications. However, as a futuristic alternative, the transgenic microalgae would reduce the dependency on food and fossil fuels in terms of energy production and efficacy.

## Figures and Tables

**Figure 1 ijms-24-01898-f001:**
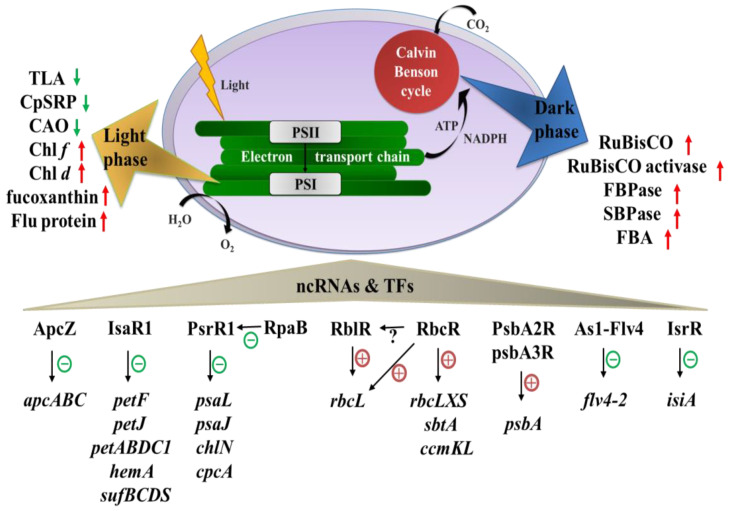
Strategies for increasing photosynthesis in microalgae. Red arrows represent overexpressed genes or exogenous genes, and green arrows represent down-regulated genes or knockout genes. Red crosses indicate an accelerative effect, and green minus signs indicate an inhibitory effect.

**Table 2 ijms-24-01898-t002:** Potential role of bio pigments from microalgae.

Pigment	Microalgae Strains	Application	Reference
*β*-carotene	*Dunaliella salina*, *Chlorella zofingiensis*, *Spirulina* spp.	The precursor of vitamin A, its antioxidant property, and its use to prevent macular degeneration, asthma, pharmaceutical, and cosmetics	[86,87]
Astaxanthin	*Haematococus pluvialis*, *Nannochloropsis oculate*, *Chlorococcus* spp.	UV protection, food colorant, anti-aging, immune enhancement, pharmaceutical, anti-hypertensive, and anti-cancer properties; anti-inflammatory	[88,89]
Lutein	*Chlorella vulgaris*, *Chlorococcum citroforme*	Feed additive and food colorant aid in the regulation of cancers, cardiovascular diseases, cognitive function, and age-related macular degeneration in humans	[90,91]
Zeaxanthin	*Nannochloropsis oculate*, *Porphyridium cruentum*	Food additives, amelioration of age-related macular degeneration, antioxidants, anti-inflammatory agents, and prevention of neurological disease	[92,93]
Fucoxanthin	*Phaeodactylum tricornutum*	Anti-cancer, anti-inflammatory, and anti-obesity effects	[94,95]
Phycocyanin	*Spirulina* spp., *Arthrospira platensis*	Used as fluorescent reagents for hepatoprotective activity, antioxidant activity, anti-inflammatory activity, and neuroprotective activity	[96,97]
Lycopene	*Chlorella marina*	Antioxidants are used as treatments for cardiovascular diseases and prostate cancer	[98,99]

## Data Availability

Not application.

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
