# Peer review of "Advances in Genetic Engineering in Improving Photosynthesis and Microalgal Productivity"

_ijms, 2023, doi:10.3390/ijms24031898_

Round 1

Reviewer 1 Report

Hu and coworkers present a well written review about the optimization of photosynthesis to enhance microalgae productivity. In general, the work is well presented even if with a low level of details; tuning photosynthesis is a long time debated topic and there are lots of attempts in different fields.

 Chapter 2 “Strategies for increasing photosynthetic efficiency” is well presented. I’d like to have the same level of detail of 2.3 also for the other two parts. Moreover, I think that the optimization of energy dissipated must be added. NPQ is a fundamental mechanism in microalgae leading to dissipate up to 80% of total energy in stress condition. Even if, the possibility to tune NPQ is still debated, several attempt to tune and disentangle the mechanism was done. Here some examples which, in my opinion must to be added to the manuscript, without detailing the argument but please insert them (Cantrell and Peers, 2017 10.1371/journal.pone.0179395; Perozeni et al 2019 10.1111/pce.13566; Barera et al 2021 10.1016/j.jbiotec.2020.12.023).

Chapter 3 is also well presented. Moreover, if we want to speak about genetic engineering to produce pigments, recently Perozeni and coworkers by overexpression endogenous bkt pseudogene were able to convert up to 80% of carotenoids into astaxanthin (Perozeni et al. 2021 10.1111/pbi.13364) and this strain showed also and higher productivity respect WT (Cazzaniga et al. 2022 10.1186/s13068-022-02173-3). I think this example perfectly fit the aim of the review.        

Typo

Reference 3 and 1 are the same

Reviewer 2 Report

Hu et al., reviewed the advances in genetic engineering in improving photosynthesis and microalgal productivity.

I have some comments below for the authors before this manuscript can be accepted for publishing.

(1), line 28, delete “eukaryotic”

(2), Line 42, delete “Diverse strategies are being implemented in this area”

(3), Line 48, “the purpose of this review is to identify the major sources of energy loss and ….” This sentence needs to be modified as I do not see “to identify the major sources of energy loss” in this review.

(4), Line 57. references are needed for this sentence.

(5), before section 2.1, it is better to describe photosynthesis, steps etc.

(6), Line 80, references are needed for this sentence.

(7), Line 89, what is LHCs?

(8), Line91, any findings in the BF4 and p71 mutants related to photosynthesis besides of the reduced antenna size?

(9), Line 106, same question for this diatom mutant

(10), Line 107-116, this paragraph needs to discuss more on the findings. It seems to me no significant findings in these studies.

(11), Line 129, references are needed for this sentence.

(12), Line 142, Delete “Besides” and use “In addition to”

(13), Line 155, references are needed for this sentence.

(14), Line 171, full name of HL

(15) 3 Transgenic microalgae for improved biomass production

In this part, the authors discussed lipids, pigments, and polysaccharides,

In general, algal biomass contains three main components: carbohydrates, proteins, and lipids.

I think the authors need to talk more about genetic engineering in algal photosynthesis to improve the production of these products which I do not see in this section.

Reviewer 3 Report

ijms-2131006-peer-review-v1 Review

This is a fairly good review article on the ways by which to improve photosynthetic productivity in microalgae. It also attempts to summarize the state of the art.  However, there are omissions and issues that authors ought to address to complete and thus improve the work.

-I strongly encourage the authors not to use the term “reduced chlorophyll antenna” or “reduced antenna.” It has caused confusion in the field as “reduction” in general, and “chlorophyll reduction” in particular is the fundamental reaction of electron transport in photosynthesis and should not be confused with a “decreased” or “truncated” antenna size. Unfortunately, “antenna reduction” has been used in some published literature to denote a “decreased” or “truncated” antenna size. However, but it has since been agreed that such misnomer needs to be corrected. Thus, authors are asked not to use the term “antenna reduction” to avoid confusion in this review. Please limit the term “reduction” to oxidation-reduction reactions only. 

Authors need to cite and discuss the first papers that demonstrated the utility of a truncated antenna size and increased biomass productivity resulting from a truncated antenna size in green algae by Polle et al. (2000 and 2003):

Polle JEW, Benemann JR, Tanaka A, Melis A (2000) Photosynthetic apparatus organization and function in wild type and a Chl b-less mutant of Chlamydomonas reinhardtii. Dependence on carbon source.  Planta 211: 335-344

Polle JEW, Kanakagiri S, Melis A (2003) tla1, a DNA insertional transformant of the green alga Chlamydomonas reinhardtii with a truncated light-harvesting chlorophyll antenna size. Planta 217: 49-59

Critical is also the review article by Melis (2009):

Melis A (2009) Solar energy conversion efficiencies in photosynthesis: minimizing the chlorophyll antennae to maximize efficiency. Plant Science 177: 272-280 DOI: 10.1016/j.plantsci.2009.06.005

The above-mentioned articles long preceded those listed as reference # [7], [8] and beyond, and they must be cited and discussed, if this review is to be correct, as these earlier publications set the stage for the expansive antenna engineering work that followed.  

2.2. The dark phase of photosynthesis” is also a misnomer, as there are no photosynthetic reactions taking place in the dark. Authors must replace this term, hare and throughout the manuscript with the term “2.2. The carbon reactions of photosynthesis.

Figure 1 schematic and associated text throughout the manuscript: The “Calvin cycle” of the carbon reactions of photosynthesis has been renamed the “Calvin Benson cycle”, not the CBB cycle. Please correct. 

In section 3 of this review, authors embarked to discuss genetic engineering of microalgae for specific product generation. If so, they should also include the cyanobacterial heterologous isoprene and β-phellandrene production work. It would help to enrich and complete this review article. 

Round 2

Reviewer 1 Report

The review in this form, thanks to the reviewer’s suggestions and the author’s work, is more accurate and complete. I think in this form can be published.

Reviewer 2 Report

This revised manuscript has been substantially improved after the modification. It can be published in the present form in my opinion. 

Reviewer 3 Report

Authors have properly updated their review manuscript to be inclusive of primary and essential work in this field and, therefore, I have no further comments.